

# Fine dust emissions from active sands at coastal Oceano Dunes, California

Yue Huang[1], Jasper F. Kok[1], Raleigh L. Martin[1], Nitzan Swet[2], Itzhak Katra[2], Thomas E. Gill[3], Richard L. Reynolds[4], Livia S. Freire[5]

[1]Department of Atmospheric and Oceanic Sciences, University of California, Los Angeles, Los Angeles, CA 90095, USA
[2]Department of Geography and Environmental Development, Ben Gurion University of the Negev, Be'er-Sheva 84105, Israel
[3]Department of Geological Sciences, University of Texas, El Paso, TX 79968, USA
[4]Department of Earth Sciences, University of Minnesota, Minneapolis, MN 55455, USA
[5]Department of Environmental Engineering, Federal University of Paraná, Curitiba, PR 80060, Brazil

*Correspondence to*: Yue Huang (hyue4@ucla.com)

**Abstract.** Sand dunes and other active sands generally have a low content of fine grains and, therefore, are not considered to be major dust sources in climate models. However, recent remote sensing studies have indicated that a surprisingly large fraction of dust storms are generated from regions covered by sand dunes, leading these studies to propose that sand dunes might be globally-relevant sources of dust. To help understand the dust emission potential of sand dunes and other active sands, we present *in situ* field measurements of dust emission under natural saltation from a coastal sand sheet at Oceano Dunes in California. We find that saltation drives substantial dust emissions from this setting. Laboratory analyses of sand samples suggest that these emissions are produced by aeolian abrasion of feldspars and removal of coatings of clay minerals on sand grains. We further find that this emitted dust is substantially finer than dust emitted from non-sandy soils and dust observed *in situ* over North Africa. As such, dust emitted from the sand sheet, and potentially from other active sands affected by similar dust emission processes, could have potent impacts on climate, the hydrological cycle, and human health. These measurements thus support the hypothesis that considerable emissions of fine dust can be generated by the reactivation of inactive dunes with accumulated clay minerals. This might occur due to future land-use changes and desertification, and is not currently represented in most climate models.

## 1 Introduction

Mineral dust aerosols produce important effects on the Earth system. Dust emission affects soil nutrient content (Neff et al., 2005; Reynolds et al., 2006) and contributes to land degradation (Ravi et al., 2010). Suspended dust influences climate and weather by modulating the radiation budget (Miller et al., 2004; Kok et al., 2017), modifying cloud microphysics (DeMott et al., 2015), and affecting the hydrological cycle (Ramanathan et al., 2001; Miller et al., 2004). Deposited dust lowers the albedo of snow and ice (Warren and Wiscombe, 1980) and contributes to oceanic primary production (Jickells et al., 2005),





terrestrial soil fertility (Chadwick et al., 1999), and the resultant carbon sequestration (Watson et al., 2000). Furthermore, dust could reduce visibility (Mahowald et al., 2007; Li et al., 2018) and cause risks to human health (Burnett et al., 2014).

Despite these critical impacts on various aspects of the Earth system, model simulations of dust have large uncertainties, particularly in both the quantity and global pattern of dust emission. The global dust emission rate reported by the Aerosol

Comparisons between Observations and Models (AeroCom) varies by a factor of eight, between ~500 to 4000 Tg yr^(-1) (Huneeus et al., 2011), although a recent study narrows that range to about a factor of three (Kok et al., 2017). On larger scales, although all models agree that North Africa is the world's largest dust source, the relative contributions of other source regions remain uncertain (Huneeus et al., 2011). On smaller scales, for instance within North Africa, substantial uncertainties in the spatial pattern of dust emission occur between models (Engelstaedter et al., 2006; Evan et al., 2015), and

even between different dust emission parameterizations within a given model (Kok et al., 2014; Evan et al., 2015). These discrepancies in predicted rates and spatial patterns of dust emission arise from uncertainties in the dust productivity of different landscape types, and from uncertainties in the physics of the emission process.

These two primary sources of uncertainties closely intertwine: the dust productivity of the land surface largely depends on its response to the physics of dust emission. For most landscapes, dust is mainly emitted through wind-blown (aeolian) sand

transport, which itself consists of two distinct processes: saltation and sandblasting (Shao, 2008). Saltation, the ballistic motion of sand particles, occurs when the shear velocity over a soil surface exceeds a threshold shear velocity (Bagnold, 1941), which depends on a range of soil and land cover properties (Owen, 1964; Marticorena and Bergametti, 1995; Bullard et al., 2011; Kok et al., 2012). After saltation starts, saltator impacts on the soil surface cause the release of dust into the air; this process is typically referred to as "sandblasting" or "saltation bombardment" (Shao, 2008). Sandblasting can emit dust in

four primary ways: (1) fragmentation of aggregates of clay-sized (<2 μm in diameter) and silt-sized (2-63 μm in diameter) particles in the soil (Kok, 2011a), (2) release of single fine particles trapped between the sand particles (referred to as "resident fines") (Bullard et al. 2004), (3) rupturing of clay- and oxide-coatings attached to the surfaces of sand grains (Bullard et al., 2004; 2007), and (4) chipping or spalling of sand grains, for instance of sharp corners, which is also known as aeolian abrasion (Kuenen, 1960; Whalley et al., 1987; Jerolmack and Brzinski, 2010; Sweeney et al., 2016; Bristow and

Moller, 2018). The relative importance of each physical process depends largely on soil texture and composition: aggregate breakage is likely most important for soils with a large fraction of fine (clay- and silt-sized) grains (Shao, 2008; Kok et al., 2012), whereas removal of mineral coatings and aeolian abrasion might be more important for predominantly sandy soils (Kuenen, 1960; Bullard et al., 2004; 2007; Jerolmack et al., 2011).

The dual uncertainties in dust emission physics and dust sources exist partly because most model parameterizations of dust

emission account only for emission through aggregate fragmentation, and not for removal of clay-mineral coatings or aeolian abrasion (Shao et al., 1993; Alfaro and Gomes, 2001; Ginoux et al., 2001; Zender et al., 2003). Many models use so-called preferential source functions to identify emissions in regions with a large content of fine grains held in soil aggregates, which



are implicitly used as a proxy for dust emissions through aggregate fragmentation (Ginoux et al., 2001; Tegen et al., 2002; Zender et al., 2003). As such, dust models weigh emissions heavily towards regions with soils that have a large proportion of fines. Because typical active sand dunes contain only a small fraction of fines (< 2% by mass of clay- and silt-sized fines; McKee, 1979), emissions from active sand dunes do not contribute substantially to dust emissions in current models (Shao et
al., 1993; Ginoux et al., 2001; Tegen et al., 2002; Zender et al., 2003).

Recent observations cast doubt on the global modeling prediction that dust emissions from active sand dunes are not a major contributor to the global dust cycle. Remote sensing observations reported a higher correlation between dust emission events and strength of winds over the dune-covered units than winds over other geomorphic units of the Sahara Desert, the Chihuahuan Desert, the Lake Eyre Basin, and the Taklamakan Area (Bullard et al., 2011; Crouvi et al., 2012). Crouvi et al.
(2012) found that more than 40% of Saharan dust storms originated from sand dunes, and that sand dunes had a higher land erodibility (regression of number of dust storms against surface wind) than other soil types. Based on these observations, Crouvi et al. (2012) proposed aeolian abrasion on active sand dunes as an important dust emission mechanism. Currently, sand dunes cover around 20% of arid regions (Pye and Tsoar, 2009), and about half of sand dunes are active, i.e., susceptible to wind erosion (Ashkenazy et al., 2012). However, changes in wind strength and rainfall, driven by climate and land-use
changes, could lead to reactivation of stable sand dunes and an increase in the coverage of active sands in the future (Yizhaq et al., 2009; Forman et al., 2001; Ashkenazy et al., 2012). Indeed, during the Late Pleistocene, sand dunes in the Sahara were mobilized then restabilized during the early Holocene, after which they were again mobilized during the late Holocene (Swezey, 2000).

Determining whether active sand dunes, and other active sands, are important global dust sources could be critical to
understanding past, present, and future dust impacts on the Earth system. However, our understanding of whether and how active sand dunes contribute to the global dust cycle is hampered by the difficulties in direct measurements and the limitations of remote sensing observations. Ninety-nine percent of the world's active sand dunes are located in remote deserts (Pye and Tsoar, 2009; Ashkenazy et al., 2012), where the installation and maintenance of equipment are difficult (Engelstaedter et al., 2006). Remote sensing observations are thus particularly informative for such regions, and these have
informed delineation of regional dust emission patterns (Prospero et al., 2002). However, on smaller scales, uncertainties in back-tracking satellite images to locate dust sources, combined with the lack of information on dust concentration, small-scale land-surface dynamics, and magnitude of wind stress, impede resolving the exact locations of dust sources (Bullard et al., 2011).

To help inform whether active sands can be regionally or globally important sources of dust emissions, we present *in situ*
field measurements of dust emission under natural aeolian saltation from active sands at the coastal Oceano Dunes State Park in California. Section 2 details the field campaign setup, *in situ* data processing methods, and laboratory techniques used to analyse sand samples. Section 3 presents results of the vertical dust mass flux, the particle size distribution of emitted dust,





the size-resolved sandblasting efficiencies, and the properties of sand samples. Section 4 discusses dust-emission processes at the study site and their representativeness for dust emissions from other active sands, and the implications of our results for climate, the hydrological cycle, and human health. Conclusions follow in Section 5.

## 2  Methods

We conducted a field campaign from May 15$^{th}$ to June 7$^{th}$, 2015 on the southern edge of the Oceano Dunes State Vehicular Recreation Area (35.03°N,120.63°W) where off-highway motor vehicles were prohibited. At the time of measurements, the site was not downwind of any vehicle activity and was completely nonvegetated and topographically mostly flat. This active sand sheet was followed inland by low transverse dunes. These dunes are part of the broader Late Quaternary Guadalupe-Nipomo Dune Complex, which extends north-south along roughly 25 kilometers of California's Central Coast and is shaped

by strong onshore (westerly) sea breezes transporting sand derived from fluvial deposits of the Santa Maria River estuary at the south end of the dune field (Cooper, 1967; Orme, 1992; Pye and Tsoar, 2009). Currently, active aeolian transport occurs only within 1-2 kilometers of the beach, but inactive, vegetation-stabilized transverse dune surfaces extend up to 18 kilometers inland.

We erected a 10-meter tower (Fig. 1) at a distance of ~650 meters from the shoreline to measure wind speed and direction,

sand flux, and dust concentration from May 15$^{th}$ to June 4$^{th}$, 2015. We mounted six Campbell Scientific CSAT3 sonic anemometers (Liu et al., 2001) on the tower, of which we used the three-dimensional winds measured by the lowest sonic anemometer (about 0.5 meters above the surface) to determine shear stress and shear velocity (details in Martin and Kok, 2017). In addition, we obtained the vertical profile of horizontal sand saltation flux $q(z)$ from nine Wenglor YH03PCT8 electric laser gate sensors (Barchyn et al., 2014), mounted at heights $z$ spanning from 0.02-0.47 meter above the surface. We

converted the measured particle count to sand flux by calibrating to eight concurrently sampling Big Spring Number Eight (BSNE) samplers (Fryrear, 1986; Goossens et al., 2000). We then obtained the total sand flux by integrating the flux through the depth of the saltation layer (see Martin and Kok, 2017; Martin et al., 2018). Furthermore, we obtained the dust concentration profile using six identical optical particle counters (OPCs) (the 212 ambient particulate profiler, manufactured by Met One Instruments, Inc.) of which we mounted four on the tower at four different heights at any given time within

0.74-6.44 meter above the surface (location details in Table S1). Each OPC measured size-resolved aerosol concentrations using seven size bins with equivalent light scattering diameter ranges within 0.49-10 μm, calibrated using polystyrene latex spheres (PSLs) (Table 1, column 1).

The optical sizing of aerosols is sensitive to differences in the refractive index between the measured aerosols and the particles used to calibrate the instrument (Hinds, 1999). The manufacturer calibrated the OPCs against PSLs, following the

international standard ISO 21501-1:2009 (ISO, 2009). We corrected the manufacturer-provided bin size limits to dust size limits using Lorenz-Mie theory (Bohren and Huffman, 1983; Mätzler, 2002), thereby approximating dust as spherical



particles. Specifically, we calculated the light scattered by PSLs ($m = 1.59 - 0i$) at the instrument's laser wavelength (589 nm) and scattering angle range ($90° \pm 60°$; information provided by Met One Engineering Department), using the manufacturer-provided size boundaries of each of the seven OPC bins. For each bin, we then determined the size range of dust particles that would produce an equal range of scattered light, using globally-representative dust refractive indexes (real part $n = 1.53 \pm 0.03$ and imaginary part $k = -10^{-2.5 \pm 0.3}$, after Kok et al., 2017). This operation yielded the corrected bin size boundaries for dust and their uncertainties (Table 1, column 2, Fig. S4 and Section 2 in the Supplement).

Using the instrumentation and procedure mentioned above, we obtained shear stress, total sand flux, and size-resolved dust concentration, which we averaged over 30-minute periods. We chose a 30-minute averaging time interval because this interval is shorter than the typical time scale of a mesoscale weather system but sufficient to capture most of the turbulent energy spectrum (Stull, 1988; Sterk et al., 1998). Furthermore, we intercalibrated the six OPCs to reduce systematic errors in the measured dust concentration and to quantify the instrument uncertainty. To obtain measurements for this intercalibration, we mounted five OPCs (the sixth one malfunctioned after May 25[th], 2015) at the same height and in a line perpendicular to the wind for three days (June 5[th], 6[th], and 7[th], 2015) immediately following the field campaign (Fig. S2). Specifically, for each of the seven size bins, we applied linear-least squares regression on the aerosol concentration of each of the five OPCs against the mean of the five OPCs (Fig. S3A). This procedure yielded a correction factor with uncertainty for each size bin of each OPC (Table S2), which we propagated throughout our analysis (Fig. S3B). For all of our subsequent analysis, we did not use dust concentrations measured by the malfunctioning OPC, because they did not satisfy our data-quality control criteria (for more details, see Section 1 in the Supplement).

We used the calibrated dust concentration to obtain the size-resolved mass flux of emitted dust using the gradient method, which is analogous to the methods for determining vertical scalar fluxes from turbulent and molecular diffusion in the atmospheric surface layer (Gillette et al., 1972; Shao, 2008). The gradient method assumes constant dust flux within the surface layer, neutral atmospheric stability (see Fig. S2 of Martin and Kok, 2017), and negligible dust deposition from upwind sources, and is expressed as

$$F_{d,i} = -K_d \frac{\partial c_i}{\partial z},$$ (1)

$$K_d = \eta \kappa u_* z,$$ (2)

where $F_{d,i}$ is the vertical mass flux (kg/m²/s) of the $i^{th}$ size class of the seven bins, $c_i$ denotes the mass concentration (kg/m³) of the $i^{th}$ size class at height $z$ (m), $K_d$ is the turbulent diffusivity (m²/s) obtained by mixing length theory (Stull, 1988), $\eta$ is the ratio between the turbulent diffusivity of a passive tracer and that of momentum, which we take as unity based on previous studies (Gillette et al., 1972; Stull, 1988), $\kappa$ is the von Kármán constant, which we take as 0.387 (Andreas





et al., 2006), and $u_*$ is the shear velocity (m/s). Combining Eqs. (1) and (2), and integrating from a reference height $z_r$ (m), we obtain

$$c_i(z) = c_i(z_r) - \frac{F_{d,i}}{\kappa u_* \eta} ln\left(\frac{z}{z_r}\right),\qquad(3)$$

where $c_i(z_r)$ is the reference concentration of the $i^{th}$ size class at $z_r$. For calculation purposes, we set $z_r$ as the height of the

"D1" OPC (Table S1 and Fig. 1). Applying Eq. (3) to each of the seven bin classes yields the size-resolved vertical dust mass flux $F_{d,i}$ for each bin. We then obtained the bulk vertical mass flux as the sum of the size-resolved fluxes $F_d = \sum_{i=1}^{7} F_{d,i}$. Note that for each bin we only used measurement intervals that showed a negative gradient (concentration decreases with height) and thus a positive dust flux, as discussed further below.

We used the size-resolved vertical dust flux obtained above to calculate the particle size distribution (PSD) of emitted dust

and the sandblasting efficiency, which together give insights into the physical processes governing dust emission from active sands (Kok et al., 2014; Mahowald et al., 2014). We computed the normalized volume PSD of dust at emission as

$$\frac{dV_i}{dlnD_i} = \frac{F_{d,i}}{F_d \cdot ln(D_{i+1}/D_i)},\qquad(4)$$

where $D_i$ (μm) and $D_{i+1}$ (μm) are the lower and upper boundary geometric diameter sizes, respectively, of the $i^{th}$ size class (Table 1, column 2). The integral of $\frac{dV_i}{dlnD_i}$ over particle size thus yields unity. For each size bin we then obtained the

sandblasting efficiency, which is the vertical dust flux produced by a unit horizontal sand saltation flux (Marticorena and Bergametti, 1995),

$$\alpha_i = \frac{F_{d,i}}{Q},\qquad(5)$$

where $Q$ (kg/m/s) is the total horizontal sand flux integrated over all sand grain sizes (see Martin and Kok, 2017). We then obtained the bulk sandblasting efficiency $\alpha$ (m$^{-1}$) by summing over the seven $\alpha_i$ (m$^{-1}$), $\alpha = \sum_{i=1}^{7} \alpha_i$.

We used the size-resolved vertical mass flux calculated above to obtain the vertical flux for particulate matter with geometric diameter $D_g \leq 10$ μm ($PM_{10,g}$) and with aerodynamic diameter $D_a \leq 2.5$ μm ($PM_{2.5,a}$). Dust in atmospheric circulation models is usually represented in terms of geometric diameter (Mahowald et al., 2014), and the $PM_{10,g}$ size range is considered most relevant to dust impacts on weather and climate (Kok et al., 2017). In contrast, aerodynamic diameter is more relevant to aerosol impacts on human health, which corresponds with the $PM_{2.5,a}$ concentration (Burnett et al., 2014).

To obtain the $PM_{10,g}$ flux, we summed the mass flux of the smallest six size bins and part of the seventh size bin that is within the $PM_{10,g}$ size range (see Table 1, column 2), for which we integrated over the sub-bin size distribution obtained by



linear-least squares regression on the PSD of the emitted dust of the sixth and seventh bin. To obtain the $PM_{2.5,a}$ flux, we first converted the geometric diameter bin sizes to aerodynamic diameter bin sizes through (Hinds, 1999)

$$D_g = \sqrt{\frac{\chi \rho_0}{\rho_P}} \, D_a \,, \tag{6}$$

where $D_a$ and $D_g$ are aerodynamic and geometric diameter, respectively, $\rho_0 = 1000 \ \text{kg/m}^3$ is the density of water, $\rho_P \approx$

$(2.5 \pm 0.2) \times 10^3 \ \text{kg/m}^3$ is the typical density of dust aerosols (Kok et al., 2017), and $\chi$ is the dynamic shape factor, which is defined as the ratio of the drag force experienced by the irregular particle to the drag force experienced by a spherical particle with diameter $D_g$ (Hinds, 1999). We used $\chi \approx 1.4 \pm 0.1$ (Kok et al., 2014), which yielded $D_g \approx (0.75 \pm 0.04)D_a$ (values of geometric and aerodynamic bin diameters are listed in Table 1). Second, we summed the mass flux of the smallest two size bins and part of the third size bin that is within the $PM_{2.5,a}$ size range (see Table 1, column 3), for which we

integrated over the sub-bin size distribution obtained by linear-least squares regression on the PSD of the emitted dust of the second and third bin.

We found that aerosol concentration, vertical mass flux, and the PSD calculated with the procedure above were affected by both dust emission and sea-salt aerosol deposition. Specifically, we found deviations in the measured aerosol concentration profiles (Fig. S5) from the logarithmic profile expected to occur from an active dust emission source (Stull, 1988; Kind,

1992; Gillies and Berkofsky, 2004), a result that we inferred as the influence of sea-salt aerosol. Because we measured dust concentrations ~650 meters from the shoreline, we expect increasing sea-salt aerosol concentration with height due to the upwind deposition of near-surface sea-salt aerosol (Liang et al., 2016). We generally observed an increasing concentration with height for the lowest two or three OPCs when saltation was inactive (horizontal saltation flux $Q = 0$), consistent with sea-salt aerosol deposition, but found a decrease in concentration with height when saltation was active ($Q > 0$), consistent

with dust emission (Fig. S5). Furthermore, the measured PSD was coarser when saltation was inactive than when it was active (Fig. S6). This observation is consistent with sea-salt aerosol being coarser than dust aerosol (O'Dowd and de Leeuw, 2007), and dominating when dust emission is not occurring. We thus mitigated the problem of the influence of sea-salt aerosol on our results by using only the lowest two sensors (D1 and D2 in Fig. 1), which were most affected by dust emission and least affected by the upwind sea-salt aerosol emission. Indeed, using only the lowest two sensors caused the

aerosol flux to be small and negative (deposition) when saltation was inactive, and large and positive (emission) when saltation was active (Fig. S7). Because using the lowest two sensors did not eliminate the deposition flux of sea-salt aerosol from our results, we in addition subtracted mass flux measured by D1 and D2 when saltation was inactive from the flux by D1 and D2 when saltation was active. Here we assumed sea-salt aerosol deposition flux to be invariant to shear velocity (Fig. S8A), because of the unrealistic decreasing trend with shear velocity in linear-least squares regression (bin 2, bin 3 and bin 4

in Fig. S8B). In addition, the assumption of the invariance of sea-salt aerosol deposition flux to shear velocity does not qualitatively affect the PSD of dust at emission (Fig. S9).





In order to characterize the sand grains at the experimental site, we collected sand samples (each ~220 grams) on October 14th, 2016 from the upper 2 centimeters of the surface at both the tower location and 100 meters upwind. We analyzed the properties of each sand sample using a series of physical and chemical techniques. First, we analyzed the particle size distribution (PSD) using the laser diffraction technique with the ANALYSETTE 22 MicroTec Plus, Fritsch (Swet and Katra,

2016), which measured particles within 0.08-2000 μm in optical diameter. We calculated the PSD of soil samples with a Fraunhofer diffraction model with a size resolution of 1 μm using MasControl software (Swet et al., 2018). Note that we did not convert this soil PSD in terms of optical diameter into geometric or aerodynamic diameter due to a lack of information in particle shape, refractive index, scattering angle range, and laser wavelength. Second, we analyzed the mineralogical composition of the sand grains using the X-ray powder diffraction (XRPD) method (Klute, 1986). Specifically, we used the

PANalytical Empyrean Powder Diffractometer equipped with position sensitive detector X'Celerator (Philips 1050/70). Data were collected in the θ/2θ geometry using Cu K$_\alpha$ radiation (λ = 1.54178Å) at 40 kV and 30 mA. Scans were run over ~15 minute-intervals in a 2θ range of 4-60° with a step equal to ~0.033° (Sommariva et al., 2014). The Reference Intensity Ratio (RIR) method was used to determine the concentrations of the crystalline components. The integral intensities of main peaks were taken for computation (Gualtieri, 1996; 2000). Third, we performed a qualitative examination of the sand grain surfaces

using Scanning Electron Microscopy (SEM) (Quanta 200, FEI) and the Energy Dispersive X-ray Spectroscopy (EDS) chemical analysis technique integrated within SEM. The high magnification (6x to 1,000,000x) of SEM enables a close analysis of the smallest dust particles (<2 μm).

## 3  Results

We find that dust emitted from the sand sheet at Oceano differs from dust emitted from non-sandy soils (defined as

containing >10% by mass of clay- and silt-sized fines; McKee, 1979) in several key ways. First, the vertical $PM_{10,g}$ and $PM_{2.5,a}$ dust fluxes at Oceano are both smaller than those fluxes from most non-sandy soils at the same shear velocity (Fig. 2). The vertical $PM_{10,g}$ and $PM_{2.5,a}$ fluxes at Oceano range from 1 to 100 μg/m$^2$/s and 0.1 to 30 μg/m$^2$/s, respectively. They both increase non-linearly with increasing shear velocity in the measured range of 0.29-0.43 m/s. For similar shear velocities, the $PM_{10,g}$ and $PM_{2.5,a}$ dust fluxes from most non-sandy soils exceed those at Oceano, differing by a factor of ~1-

100 and ~0.1-10, respectively. The second key difference is that dust emitted from the Oceano site is substantially finer than size-resolved dust emitted from non-sandy soils under natural saltation (Fig. 3A), and it is also significantly finer than dust measured *in situ* over North Africa (Fig. 3B), which likely accounts for a majority of the world's dust emission (Prospero et al., 2002; Engelstaedter et al., 2006). The third key difference is the dependence of the emitted dust particle size distribution (PSD) on shear velocity. The PSD of dust at emission at Oceano shifts to finer size with increasing shear velocity (Fig. 4).

Specifically, the fraction of emitted fine dust ($D_g < 4$ μm) increases with increasing shear velocity, whereas the fraction of coarse dust decreases substantially. In contrast, the PSDs of dust emitted from non-sandy soils appear invariant to shear velocity in previous studies (Fratini et al., 2007; Kok, 2011b; Shao et al., 2011; Mahowald et al., 2014).





Our measurements of the size-resolved sandblasting efficiency provide further insight into the differences between dust emissions from the Oceano sand sheet and from non-sandy soils. First, the bulk sandblasting efficiency at Oceano is around $10^{-6}$ m$^{-1}$ (Fig. 5), which is substantially smaller than the range within $10^{-5}$ to $10^{-2}$ m$^{-1}$ typical for non-sandy soils (Kok et al., 2012). Second, the bulk sandblasting efficiency increases non-linearly as a power law in shear velocity (Fig. 5), a
result consistent with some previous studies (Shao et al., 1993; Marticorena and Bergametti, 1995; Kok et al., 2012). However, the dependence of sandblasting efficiency on shear velocity changes with dust size (Fig. 6): whereas the power exponent increases with dust size from 4.7 to 6.1 for the smaller four bins, the exponent decreases from 3.3 to -8.5 for the larger three bins. This transition from large and positive exponents for fine dust to small and negative exponents for coarse dust drives the shift towards a finer dust PSD that we observed with increasing shear velocity (Fig. 4), which does not occur
for non-sandy soils (Kok 2011b; Shao et al., 2011).

Our analyses of sand grains sampled from the field site show that the Oceano sand is characterized by a unimodal size distribution (mode at 461 μm and median at 491 μm) with 0.95% (by mass) of loose clay- and silt-sized fine (<63 μm) between the sand grains and 0.41% of particulate matter smaller than 10 μm (Fig. 7). The sand consists of a mixture of quartz (51% by mass), feldspars (K-rich feldspar 23% and plagioclase 23%), and clay minerals (3%) analyzed by the XRPD
technique (Table 2). We further find the presence of mineral coatings on the top of quartz sand grains through the SEM-EDS technique (Fig. 8D and 8E). Note that the mass content of clay minerals detected in the XRPD analysis can be in the form of loose fine dust in the pore spaces between the sand grains and/or clay-mineral coatings attached to the sand grains. In addition, we find that the surface of the feldspars appears more abraded than the solid surface of the quartz sand grains (compare location "B" and location "C" in Fig. 8A), which are representative of a large number of SEM-EDS observations
made for these samples.

## 4  Discussion

We reported *in situ* field measurements of natural dust emission from an undisturbed coastal sand sheet at Oceano Dunes State Park in California (Figs. 1 and S1). We found that dust emission from these active sands differs in several key ways from dust emission from non-sandy soils, namely in the magnitude of vertical PM$_{10,g}$ and PM$_{2.5,a}$ dust fluxes (Fig. 2), the
particle size distribution (PSD) of dust at emission (Fig. 3), the shear velocity dependence of the emitted dust PSD (Fig. 4), and the magnitude and the shear velocity dependence of size-resolved sandblasting efficiency (Figs. 5 and 6). Furthermore, we found that sand grains at the study site are coarse with a mode larger than 460 μm (Fig. 7), that many sand grains contain clay-mineral coatings, and that about half of the sand grains are feldspars (K-rich feldspar and plagioclase) (Table 2). These results provide insights into several fundamental questions: (1) what physical processes drive dust emission from active
Oceano sands? (2) how representative are dust emissions from the sand sheet at Oceano of active sands elsewhere? and (3)





what are the implications of dust emission from Oceano for climate, the hydrological cycle, human health, and park management? After addressing these questions, we end the discussion section with several limitations of our methodology.

### 4.1 Insights into processes producing dust from active sands at Oceano Dunes State Park

Our results indicate that aeolian abrasion of feldspars is one possible dust emission process at Oceano. XRPD analysis confirms the existence of feldspars (~46% by mass) (Table 2). Although the content of feldspars is of the same magnitude as that of quartz (~51% by mass) (Table 2), the surfaces of the feldspars commonly appear more abraded than the surfaces of the quartz sand grains (Fig. 8A), suggesting a higher potential to generate dust through aeolian abrasion of feldspars than of quartz grains. We list four supporting findings from other experimental studies. First, both laboratory (Dutta et al., 1993; Jari, 1995) and wind tunnel (Kuenen, 1960; 1969) experiments have found that feldspars are more fragile than quartz grains to aeolian abrasion, possibly owing to the cleavage structures of feldspars (Kuenen, 1969) and the greater propensity of feldspars to weather than quartz (Nesbitt et al., 1997). Second, the feldspar content of dust is higher than that of the parent top soils in dust-producing regions, including Northern Ghana (Tiessen et al., 1991), China (Feng et al., 2008), and the UK (Moreno et al., 2003), implying the selective removal of feldspars to generate dust. Third, laboratory experiments have found that aeolian abrasion of feldspars can generate a larger fraction of the $PM_{10}$ and the $PM_{2.5}$ size ranges than generated from abrasion of quartz (Domingo et al., 2010). Indeed, laboratory experiments on washed and clean quartz grains (for which grain surface minerals have been removed) (Whalley et al., 1987; Bullard et al., 2007) and on freshly crushed quartz grains (Wright et al., 1998; Wright, 2001) imply that the dust produced by aeolian abrasion has a small portion (less than 1% by mass) within $PM_{10}$ size range. Fourth, Pye and Sperling (1983) found that when coastal dune sands are exposed to salt solutions under desert diurnal temperature and humidity cycles, feldspars are more susceptible to breakage into fine grains than quartz. Because the Oceano measurement site is exposed to high levels of sea-salt aerosols deposition (see Methods), this finding further supports our hypothesis that aeolian abrasion of feldspars is one possible dust emission process at Oceano.

Our results further suggest that removal of clay-mineral coatings by saltator impact is another possible contributor to dust emissions at Oceano. First, XRPD analysis found the content of clay minerals to be around 3% by mass (Table 2) for Oceano sand samples, although the fraction of this contributed by clay-mineral coatings is uncertain because XRPD analysis cannot distinguish between clay-mineral coatings and loose clay-sized fines trapped in the pore spaces between sand grains (Swet et al., 2018). However, the PSD of dust emitted from the Oceano site peaks around 2 μm in diameter, which is much finer than emitted from non-sandy soils (Figs. 3 and 4). This PSD is generally consistent with the PSD of emitted dust of laboratory experiments on sand samples taken from Australian active sand dunes (Bullard et al., 2004; 2007). Specifically, Bullard and co-workers simulated aeolian abrasion within an air chamber on the Australian sand grains with the content of clay-mineral coatings as ~1% - 2% by mass. They concluded that the removal of clay-mineral coatings was likely the main dust emission process, implying that removal of clay-mineral coatings is a key emission process at Oceano as well. Second, a companion





paper reports results of wind tunnel experiments on sand grains collected from our Oceano field site, as well as from two active desert dune fields in Israel (Swet et al., 2018). They reported that all three sands produced substantial dust emissions, and their measured sandblasting efficiency of Oceano sands was consistent with our field measurements. After Swet and co-workers washed the sand grains from the two Israeli dune fields to remove most of the loose resident fines, leaving the clay-mineral coatings remained largely unchanged, they found that the washed sand grains still produced $PM_{10}$ dust emissions that were comparable to those before the sand was washed. Although Swet et al. (2018) did not perform this last experiment on Oceano sand, these results support the interpretation that removal of clay-mineral coatings is a key process driving dust emissions from active sands, including at Oceano.

Our results imply that sandblasting of resident fines by saltators is a third possible dust emission process at Oceano. The resident fines exist as single particles trapped in between the sand grains, as was observed when we washed sand samples in the lab. Although XRPD analysis cannot distinguish between clay-mineral coatings and clay-sized resident fines in the pore spaces between sand grains (Swet et al., 2018), the content of these resident fines is less than 3% by mass (Table 2). Because we observed a net dust emission flux only when saltation was occurring (Figs. S4 and S6), it is unlikely that the direct aerodynamic entrainment of resident fines (Klose and Shao, 2012) contributed substantially to dust emissions. The occurrence of dust emission by removal of resident fines is supported by the measurements of Swet et al. (2018), who reported that unwashed sands produced more dust than washed ones.

Our results provide several insights into the energetics of dust emission from Oceano. First, the lower bulk sandblasting efficiency at Oceano, compared to that of non-sandy soils, supports the common hypothesis that sandblasting efficiency increases sharply with clay-sized grain content (Marticorena and Bergametti, 1995; Kok et al., 2014). Second, we find that the bulk sandblasting efficiency of Oceano sand increases as a power law with shear velocity (Fig. 5). Because such enhancement of sandblasting efficiency occurs despite invariance of mean saltator velocity with shear velocity (Martin and Kok, 2017), such changes in sandblasting efficiency must be driven instead by changes in the probability distribution of the saltator impact energies. The recent dust-emission model of Kok et al. (2014) predicts a power law increase of the sandblasting efficiency with shear velocity for soils for which the typical saltator impact energy is substantially less than the threshold impact energy needed to overcome particle bonds. This power law increase occurs because, for such erosion-resistant soils, only particularly energetic saltator impacts are capable of emitting dust, and the fraction of saltators impacts that are particularly energetic increases non-linearly with wind shear velocity (see Fig. 16 in Kok et al., 2012; Fig.1 in Kok et al., 2014). As such, our results both tentatively support the Kok et al. (2014) dust-emission model and suggest that dust emission from Oceano sands are predominantly produced by saltators with an impact energy much greater than the mean impact energy (i.e., with impact energy in the high-energy tail of the saltator-impact-energy distribution). This dependence of dust emission on the very energetic saltator impacts likely occurs because both removal of clay-mineral coatings and aeolian abrasion of feldspars are energetic processes, and thus require particularly energetic saltator impacts. The third



insight into the energetics of dust emission arises from the size-resolved sandblasting efficiency in Fig. 6, which shows a clear transition of the dependence on shear velocity from the smaller size bins to the larger ones. Specifically, the power exponents are large and positive for fine dust (up to bin 5 and $D_g = \sim 6$ μm), and transition to small and negative exponents for coarse dust (bins 6 and 7 and $D_g > \sim 6$ μm). This transition drives the shift towards a finer dust PSD that we observed

with increasing shear velocity (Fig. 4). This shift towards finer dust could similarly be due to the increase in the proportion of saltators with very high impact energies that occurs with increasing shear velocity (Kok et al., 2012; 2014). The resulting increasing energy of the saltator impacts that emit dust could either cause some of the emitted coarser dust to get fragmented into smaller dust particles, or could strip more energetically-bound fine particles off the surfaces of sand grains (Alfaro and Gomes, 2001).

**4.2   Insights into representativeness of Oceano dust emissions for other active sands**

It is unclear how representative our measurements on the active coastal sand sheet at Oceano are of dust emission from other active sands, including active sand dunes, in particular in relation to the sand size distribution, the content of feldspars, and the extensiveness of clay-mineral coatings.

Our results indicate that sand at our study site, with a mode around 461 μm (Fig. 7) is coarser than sand of most active sand

dunes (Bullard et al., 2004; Pye and Tsoar, 2009; Webb et al., 2013; Swet et al., 2018). Because the threshold shear velocity required to sustain saltation increases with the grain particle size (Shao, 2008; Kok et al., 2012), the value of 0.28 m/s we found at Oceano (Martin and Kok, 2017) is likely somewhat higher than that at typical active sand dunes, as also confirmed in the laboratory measurements of Swet et al. (2018). As such, the horizontal saltation flux at Oceano could be somewhat lower compared with typical active sand dunes.

Our results further indicate that Oceano sand differs from sand of typical active sand dunes in its mineralogy, specifically in the content of feldspars. Oceano sands contain as much as ~46% feldspar by mass (Table 2). This content is approximately twice the proportion of feldspar found in coastal dunes along Monterey Bay to the north (Combellick and Osborne, 1977) and more than three times as much as in coastal dunes in Baja California (Kasper-Zubillaga et al., 2007). This mineralogically immature sand suggests rapid erosion of a feldspar-rich source rock, potentially derived from the Santa

Maria River drainage. More mineralogically mature terrestrial dune fields, including most old and continental-based (as opposed to coastal) terrestrial desert sand seas and some older coastal dunes, are quartz-dominated and feldspar-depleted (Brookfield and Ahlbrandt, 1983; Pye and Tsoar, 2009; Swet et al., 2018). The removal of feldspars by ballistic-impact-derived breakdown into dust during aeolian recycling over long periods of time contributes to the mineralogical maturation (into quartz-dominated deposits) of aeolian dune sands (Muhs et al., 1997; Muhs, 2004).



Furthermore, weathering of feldspars to clay minerals might produce some of the clay-mineral coatings on sand grains (O'Hara-Dhand et al., 2010) at the study site. Feldspars in the Oceano sand, especially plagioclase, are vulnerable to chemical weathering under the humid and salt-bearing coastal conditions (James et al., 1981; Pye and Tsoar, 2009) and mechanical breakdown (aeolian abrasion) into dust from saltator impacts in wind events (Muhs et al., 1997). Additionally,
Compton (1991) described how the Santa Maria basin source sediments, both onshore and offshore, contained very fine clays derived from the alteration of volcanic glass mediated by and enhanced by weathering of feldspars. As a result, the combination of abundant feldspars in the Oceano sand and clay-minerals from an additional source could help explain the observed clay-mineral coatings on Oceano sand and the fine-grained characteristics of Oceano dust emissions.

Our results also imply that the dust-emission processes at Oceano are not main contributors to dust emissions from North
Africa. *In situ* measurements of the PSDs of atmospheric dust over North Africa are significantly coarser than the PSD of dust emitted from our Oceano site (Fig. 3B) and the PSD of dust generated in laboratory experiments on the removal of clay-mineral coatings (Bullard et al., 2004; 2007). As such, although the processes responsible for dust emission from the Oceano sand sheet - removal of clay-mineral coatings and aeolian abrasion of feldspars - could be important for specific North African dust sources, they are likely not primary dust emission processes on a regional scale in North Africa.

**4.3   Implications of dust emissions from Oceano Dunes State Vehicular Recreation Area for human health, park management, the hydrological cycle, and climate**

Dust emitted from our Oceano site is significantly finer than dust emitted from non-sandy soils (Fig. 3A), which can amplify its impacts on downwind climate, the hydrological cycle, and human health. Partially because fine dust has a longer lifetime and larger surface area per unit mass than coarse dust, it produces a substantial cooling radiative effect, which has important
implications for climate and weather (Kok et al., 2017). Moreover, $PM_{2.5,a}$ is commonly associated with cardiopulmonary diseases, lung cancer and ischemic heart disease (Burnett et al., 2014). On a per mass basis, dust emitted from Oceano, and from other source regions with similar emission processes, can thus be expected to have a greater effect on downwind climate, weather, and human health than dust emitted from non-sandy soils.

Our measurements of dust emission from the Oceano Dunes State Vehicular Recreation Area (SVRA) could help inform
decisions on the management of this California state park. Our measurements were performed in the part of the park where public off-highway vehicle (OHV) use is prohibited (http://ohv.parks.ca.gov/?page_id=1208), and we found that substantial dust emissions occur in the absence of OHV use (Fig. S1). Because OHV use can enhance dust emissions (Goossens and Buck, 2011; Goossens et al., 2012), reconfiguring motorized access to the park could modify dust emissions and associated downwind dust impacts on human health, vegetation, and water quality (Ouren et al., 2007). Disentangling the relative
contributions of natural and OHV-influenced dust emissions will be essential as Oceano Dunes SVRA responds to a dust emission abatement order recently issued by the local air quality control board (Vaughan, 2018).





Dust emitted from Oceano sands possibly contain a large proportion of feldspars, which could affect downwind cloud properties and the hydrological cycle. Oceano sand has ~46% feldspars by mass, which is on the high end for dust-source regions (Murray et al., 2012; Atkinson et al., 2013). Consequently, the content of feldspars in dust emitted from Oceano is likely on the high end as well, especially considering that aeolian abrasion of feldspars (Muhs et al. 1997) might be one of

the emission processes. Feldspars, especially K-rich feldspars, are considered the most important ice nuclei for mixed-phase clouds, based on both laboratory (Atkinson et al., 2013) and *in situ* measurements (Price et al., 2018). Recent observations suggest that cloud glaciation mediated by dust aerosols contributes to more than half of ice-phase precipitation in the Sierra Nevada mountain range (Creamean et al., 2013), which is a major source of water for Californian residents (Dettinger et al., 2004). Because Oceano sand is rich in feldspars, dust emitted from Oceano (and other dust-emitting coastal or desert dunes

with a high feldspar content), could potentially contribute to glaciating downwind clouds, thereby affecting the local and regional hydrological cycle. Considering these potentially important impacts, there is a clear need for further in-depth analysis of the feldspar abrasion mechanism. Further measurements to determine the broader occurrence of dust production by feldspar abrasion for active sands with high feldspar content are clearly needed.

The fine dust flux produced by beach sand at Oceano (Fig. 2) tentatively supports the hypothesis that dust emissions from

active sands might have been globally important sources of dust in past climates, and that emissions from active sands might increase in the future because of the reactivation of inactive sand dunes (Mason et al., 2003; Thomas et al. 2005; Bhattachan et al., 2012). Many stable dunes have accumulated extensive clay-mineral coatings (Bowler, 1973; Gardner and Pye, 1981; Muhs et al., 1997), and, if activated, these dunes could produce substantial fine dust emissions through sandblasting-induced removal of clay-mineral coatings (Bullard et al., 2007; Swet et al., 2018). The transition from stable to active sand dunes can

be triggered by increases in wind speed or decreases in vegetation coverage, which in turn can be caused by decreased precipitation or by human activities, such as grazing and land-use change (Ashkenazy et al., 2012). Areas with coexisting active and stable sand dunes, such as coastal zones, are especially vulnerable to the reactivation of stable sand dunes (Yizhaq et al., 2009), and could thus become potent dust emission sources, especially if the dunes experienced some weathering in the humid, salt-rich coastal environment (Muhs et al., 1997). Although it is unclear how representative our results are for

typical active sands, the PSD of dust at emission at Oceano is consistent with fine dust emitted from Australian active sand dunes (Bullard et al., 2004; 2007). As such, our measurements suggest that newly activated dunes might become an important source of especially fine dust. Because active sands are generally not represented as dust emission sources in climate models, reactivation of sand dunes might thus enhance future dust effects on climate beyond what current models simulate (Kok et al., 2018).

## 4.4 Limitations of the methodology

Our methodology has important limitations. First, it is uncertain how well our measurements from the active coastal Oceano sand sheet represent the process of dust emission from other active sands. Although our measured PSD of dust at emission is





consistent with laboratory measurements of dust emissions through removal of clay-mineral coatings (Bullard et al., 2007), more field measurements from active sands are needed to better quantify the flux and the PSD of emitted dust. In particular, the PSD and the flux of dust at emission produced by active sand dunes are likely to depend on various factors, including sand size distribution, mineralogy, chemical and physical weathering rates, dune type, saltation activity level, wind intensity,

threshold shear velocity, and palaeoenvironmental history. As such, our conclusions on the contribution of active sand dunes to past, current, and future dust emissions should be seen as tentative. Second, our XRPD analysis could not distinguish the concentrations of clay-mineral coatings from loose individual fine particles contained in the pore spaces between sand grains. This uncertainty limits our interpretation of the relative contributions of removal of clay-mineral coatings and saltation-driven release of loose fines to our measured dust fluxes. Third, we collected our soil samples on October 14[th],

2016, more than a year after the Oceano field campaign in the summer of 2015, such that it is possible that these samples are not representative of the sand surface at the time of our field campaign. However, sand-surface properties over a large area are unlikely to change substantially within a timescale of one year (Hillel, 1998). Fourth, we approximated dust as spherical particles when correcting the OPC bin sizes. However, dust is highly aspherical (Okada et al., 2001; Kandler et al., 2007), and the extinction efficiency of aspherical dust with $D_g \geq 1$ µm is ~20-60% larger than that of spherical dust of the same

volume (Kok et al., 2017). Consequently, our method to determine geometric diameters of dust that produce the same scattering intensities as those of PSLs likely overestimates the dust geometric diameter, especially for dust larger than 1 µm. This possibility reinforces our conclusion that dust emitted from Oceano, and possibly from active sands in general, is substantially finer than dust emitted from non-sandy soils.

## 5 Conclusion

We presented *in situ* field measurements of dust emissions under natural saltation from an active sand sheet at coastal Oceano Dunes State Park in California (Figs. 1 and S1). We found that although the $PM_{10,g}$ dust emitted from Oceano is substantially less than that emitted from non-sandy soils (Fig. 2A), it is also substantially finer (Fig. 3A), such that the $PM_{2.5,a}$ flux is not much smaller than emitted from non-sandy soils for a given shear velocity (Fig. 2B). As such, dust emissions from Oceano, and possibly from other active sands with similar emission processes, could impact downwind

climate, the hydrological cycle, and human health. Furthermore, unlike for emissions from non-sandy soils, the PSD of emitted Oceano dust shifts towards finer dust size with increasing shear velocity (Fig. 4). Finally, although the sandblasting efficiency at Oceano is smaller than for non-sandy soils, it increases non-linearly with wind shear velocity (Figs. 5 and 6).

Our results provide insights into the physical processes that drive dust emissions from the sand sheet at Oceano Dunes, and possibly from other active sands. We find that Oceano sand has substantial clay-mineral coatings and feldspars (Table 2),

suggesting that dust is emitted through a combination of removal of clay-mineral coatings, release of resident fines, and ballistic breakdown of feldspars by saltation impacts. However, it remains unclear how representative our measurements are



of dust emissions from typical active sands, considering that sand at our Oceano site was relatively coarse and immature in mineralogy.

Our measurements provide limited insights into the contribution of active sands, including active sand dunes, to the global dust cycle. We find that the PSD of dust emitted from Oceano is much finer than observed *in situ* over North Africa (Fig. 3B). Therefore, although the processes responsible for dust emission at our Oceano site - removal of clay-mineral coatings and aeolian abrasion of feldspars - could be important for specific North African dust sources, they are likely not primary dust-emission processes on a regional scale in North Africa. However, because it is unclear how representative the dust-emission processes at our Oceano site are for emission processes occurring at typical active sands, the contribution of active sands to dust emission from North Africa and other globally-important source regions remains uncertain. Further work on dust emissions from active sands with various properties is thus needed to evaluate the contribution of active sands to the global dust cycle. Nonetheless, our measurements support previous work indicating that clay-coating removal is likely a major emission process for active sands (Bullard et al., 2007). As such, dust emissions from active sands could increase because land-use changes and desertification might reactivate currently stable sand dunes, which can have accumulated extensive clay-mineral coatings (Thomas et al., 2005). Because this process is not accounted for in most current climate models, it could enhance future dust effects on climate beyond what current models simulate (Kok et al., 2018).

## Acknowledgements

We acknowledge support from National Science Foundation (NSF) grant AGS-1358621 to J.F.K., NSF Postdoctoral Fellowship EAR-1249918 to R.L.M, United States-Israel Binational Science Foundation (BSF) grant 2014178 to J.F.K. and I.K., and Brazilian National Council for Scientific and Technological Development (CNPq) fellowship to L.S.F.. We thank Hezi Yizhaq, Marcelo Chamecki, Francis A. Turney, Jack A. Gillies, Vicken Etyemezian, Richard Langford, and Cenlin He for helpful comments and discussions that helped us to improve the quality and clarity of our manuscript. We also thank Peter Rowland and Tim Pesce at Oceano Dunes State Vehicular Recreation Area for access and transportation support to the Oceano field site. Data will be deposited in a publicly-accessible repository upon paper acceptance and are available upon request during the review process.

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



**Table 1: The dust geometric and aerodynamic diameters of the eight bin boundaries.** Determination of the geometric diameter ranges uses Lorenz-Mie theory such that they produce the same range of scattered light intensity as the eight manufacturer-provided polystyrene latex sphere (PSL) diameter sizes. The ranges in geometric diameters derive from the uncertainty in the dust refractive indices. We
5    determined the aerodynamic diameter ranges by applying Eq. (6) to the geometric diameter ranges.

| Polystyrene latex sphere (PSL) diameter (μm) | Dust geometric diameter (μm) | Dust aerodynamic diameter (μm) |
|---|---|---|
| 0.49 | 0.51 – 0.54 | 0.68 – 0.72 |
| 0.7 | 0.73 – 0.77 | 0.97 – 1.03 |
| 1 | 1.01 – 1.03 | 1.35 – 1.37 |
| 2 | 2.18 – 2.32 | 2.91 – 3.09 |
| 2.5 | 2.67 – 2.90 | 3.56 – 3.87 |
| 5 | 6.11 – 6.31 | 8.15 – 8.41 |
| 7 | 8.89 – 9.40 | 11.85 – 12.53 |
| 10 | 15.67 – 16.01 | 20.89 – 21.33 |

**Table 2: Averaged mineralogy of the Oceano Dunes sand samples in unit of percentage by mass, determined from X-ray powder diffraction (XRPD) analysis.**

|  | Quartz | Potassium feldspar (Microcline and Orthoclase) | Sodium-rich plagioclase feldspar (Albite) | Clay minerals (Illite) |
|---|---|---|---|---|
| Oceano Dunes | 51 | 23 | 23 | 3 |



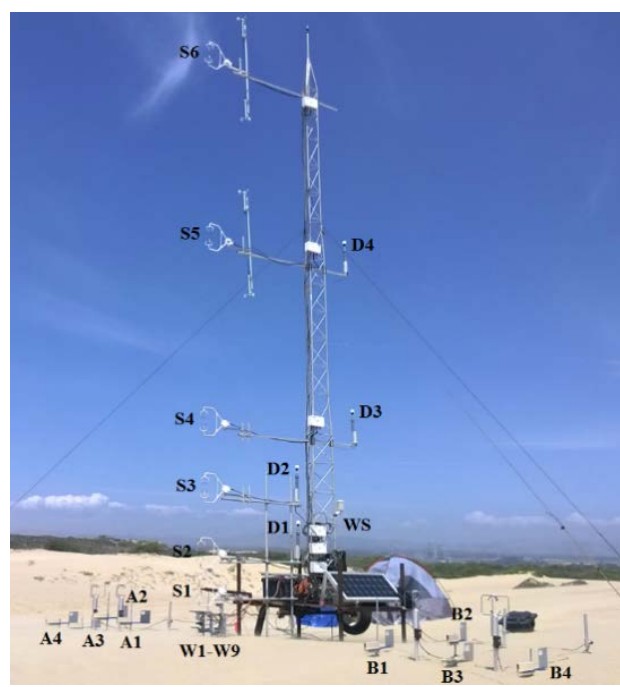

**Figure 1: The experimental setup at Oceano Dunes.** Instrumentation includes six sonic anemometers (S1-S6), nine Wenglor particle counters (W1-W9), eight Big Spring Number Eight sand samplers (A1-A4, B1-B4), four optical particle counters (D1-D4), and a weather station with temperature and humidity sensors (WS) (after Martin and Kok, 2017).

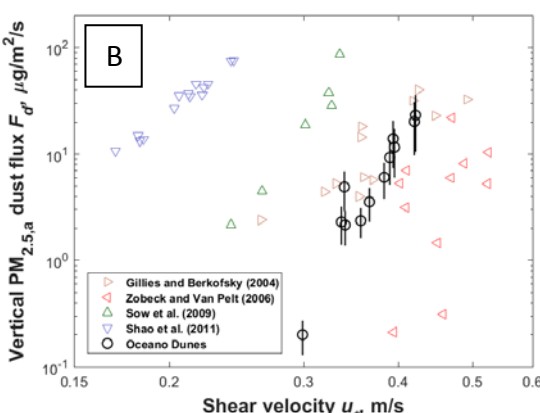

**Figure 2: The vertical dust mass flux as a function of shear velocity within (A) the PM$_{10,g}$ geometric diameter range and (B) the PM$_{2.5,a}$ aerodynamic diameter range.** Each of the two plots includes measurements at Oceano (black open circles), and previously published field studies of natural dust emissions from non-sandy soils (open triangles). These latter measurements were compiled in Kok
10  et al. (2014), which corrected these measurements to the PM$_{10,g}$ geometric diameter range following the procedure described in that work. Furthermore, we corrected these measurements to the PM$_{2.5,a}$ aerodynamic diameter range assuming that their PSDs follow the prediction of brittle fragmentation theory generated by aggregate fragmentation (Kok, 2011a). Error bars on the Oceano measurements were obtained through error propagation (details in Supplement).



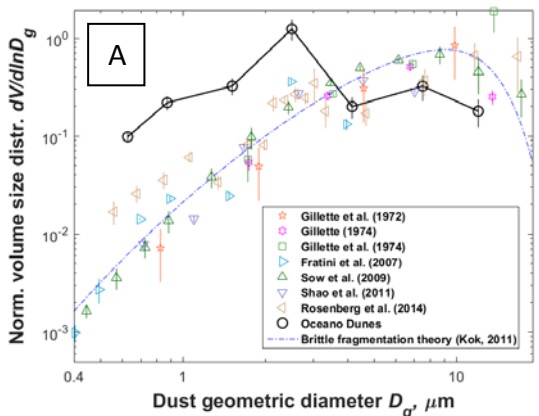
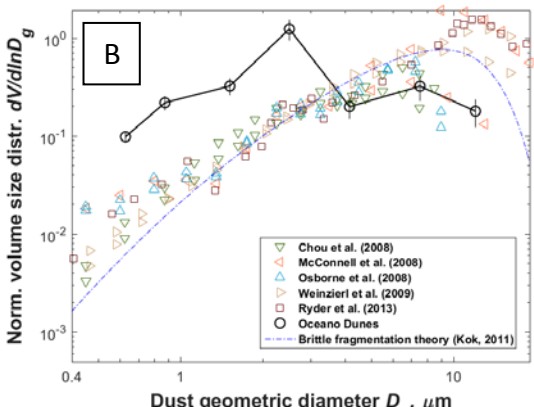

**Figure 3: Normalized volume particle size distribution (PSD) of dust at emission as a function of dust aerosol geometric diameter averaged over all measurements at Oceano (black open circles).** Plotted for comparison are measurements compiled by Mahowald et
5  al. (2014) of (A) the emitted dust PSDs from non-sandy soils in the U.S. (Gillette et al., 1972; 1974; Gillette, 1974), China (Fratini et al., 2007), North Africa (Sow et al., 2009; Rosenberg et al., 2014), and Australia (Shao et al., 2011), and (B) *in situ* aircraft measurements of dust-dominated PSDs close to the Saharan source regions. Note that these measurements are normalized using the procedure described in Kok (2011a) and Mahowald et al. (2014), which differs somewhat from the procedure used for the Oceano data (see Methods). Also plotted for comparison is the brittle fragmentation theory (blue dash-dotted lines) on the PSD of emitted dust generated by aggregate
10 fragmentation (Kok, 2011a). Error bars on the Oceano measurements obtained through error propagation (details in Supplement).

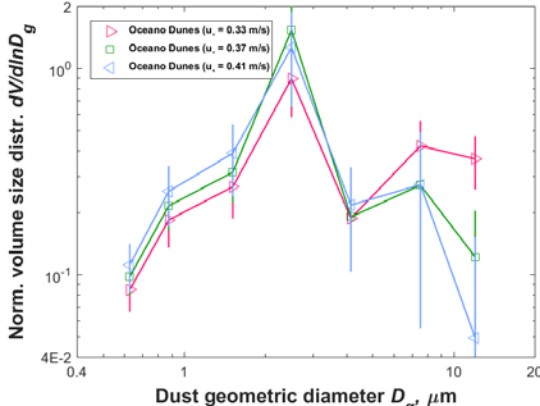

15  **Figure 4: Shear velocity dependence of the normalized volume particle size distribution of dust at emission at Oceano.** We divided the measurements evenly into three shear velocity bins, with averaged values of 0.33, 0.37 and 0.41 m/s, respectively. Error bars on the measurements obtained through error propagation (details in Supplement).



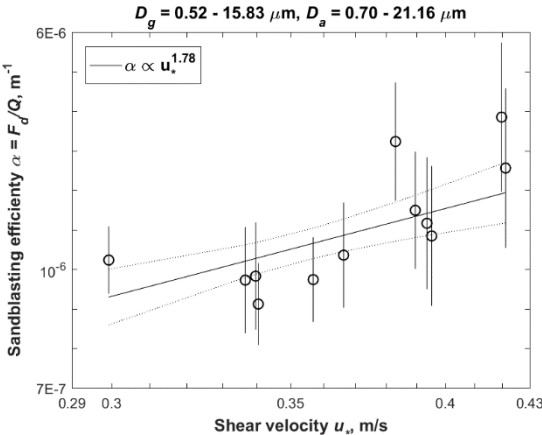

**Figure 5: The bulk sandblasting efficiency $\alpha$ $(m^{-1})$ at Oceano as a function of shear velocity.** The linear-least squares fit (black line) indicates that $\alpha$ increases as a power law in shear velocity with an exponent of 1.78±0.85. Error bars were obtained by propagating errors from the sensor intercalibration process, the regression used in the gradient method, and the subtraction of the sea-salt deposition flux (see Supplement). The two black dotted lines denote the 95% confidence range on the fit. The geometric and the aerodynamic diameter ranges of the vertical dust mass flux $F_d$ are noted at the top, and $Q$ is the total horizontal sand flux integrated over all sand grain sizes (Martin et al., 2018).

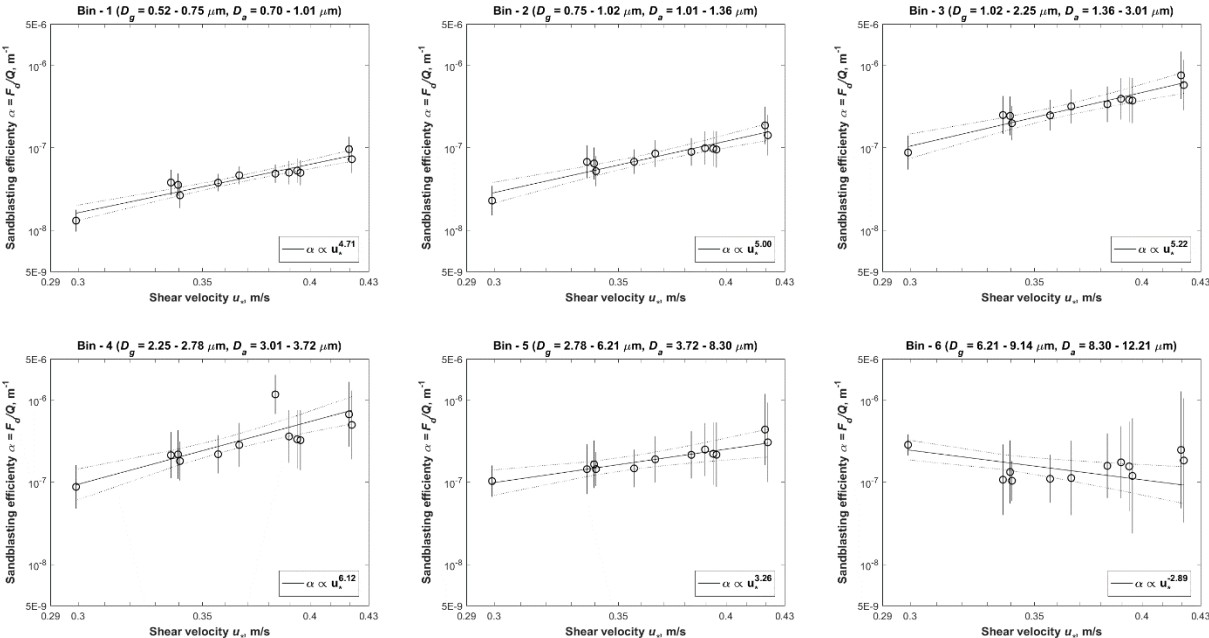

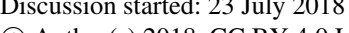



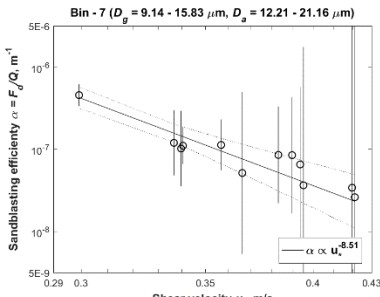

**Figure 6: The sandblasting efficiency $\alpha_i$ $\left(\mathrm{m}^{-1}\right)$ of the $i^{th}$ of the seven size bins at Oceano as a function of shear velocity.** The linear-least squares fit (black line) indicates that $\alpha_i$ varies as a power law in shear velocity with an exponent of 4.71±0.97, 5.00±1.37, 5.22±1.63, 6.12±2.11, 3.26±1.90, -2.89±1.94, -8.51±2.67, for the seven respective size bins. Error bars were obtained by propagating errors from the sensor intercalibration process, the regression used in the gradient method, and the subtraction of the sea-salt deposition flux (see Supplement). The two black dotted lines denote the 95% confidence range of the fit. The geometric and the aerodynamic diameter ranges of the vertical dust mass flux $F_{d,i}$ are noted at the top of each of the seven plots, and $Q$ is the total horizontal sand flux integrated over all sand grain sizes (Martin et al., 2018).

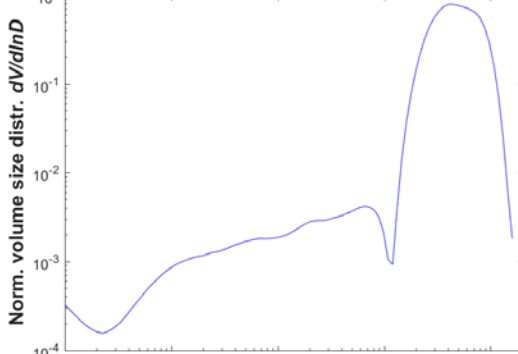

|  | Oceano |
|---|---|
| **Statistical parameters (μm)** | |
| Mean | 538.28 |
| Mode | 461.48 |
| D10 | 270.61 |
| D50 | 490.95 |
| D90 | 879.67 |
| **Weight fraction (%)** | |
| Clay-sized (< 2 μm) | 0.16 |
| Clay- and Silt-sized (< 63 μm) | 0.95 |
| Fine sand (63-250 μm) | 7.93 |
| Meidum sand (250-500 μm) | 54.35 |
| Coarse sand (> 500 μm) | 44.69 |
| | |
| $PM_1$ (< 1 μm) | 0.09 |
| $PM_{2.5}$ (< 2.5 μm) | 0.19 |
| $PM_{10}$ (< 10 μm) | 0.41 |

**Figure 7: Normalized volume particle size distribution (PSD) of sand samples collected from the Oceano field site.** Statistical parameters of the PSD are reported in the table to the right. D10, D50 and D90 refer to cut-off diameters of the 10[th]-, 50[th]-, and 90[th]-percentile of sample grains by mass, respectively.





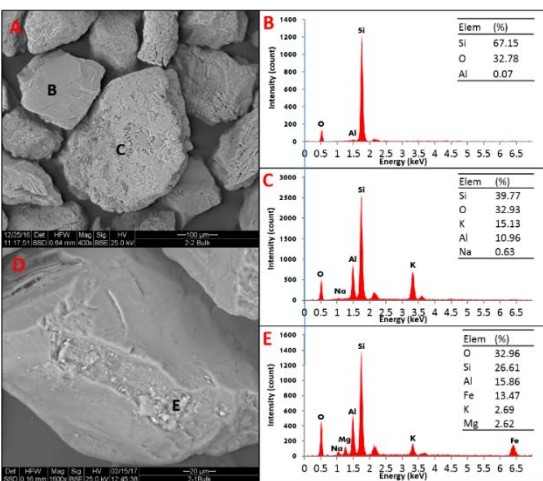

5    **Figure 8: (A) Scanning Electron Microscopy (SEM) image of an Oceano sand sample. (B) Energy Dispersive X-ray Spectroscopy (EDS) chemical composition (percentage by mass) of a quartz grain in image A (the location of the analysis is marked by the black letter "B"). (C) EDS chemical composition of a K-rich feldspar in image A marked by "C". (D) Close-up image of a quartz sand grain with mineral coatings. (E) EDS chemical composition of the mineral coatings in image D marked by "E".**

