# Peer review of "Fine dust emissions from active sands at coastal Oceano Dunes, California"

_Atmospheric Chemistry and Physics, 2018_

## Referee Comment (RC1) · V. Etyemezian (Referee) · 22 Aug 2018

Review for Fine Dust Emissions from Active Sands at Coastal Oceano Dunes, California ACP-2018-692

This paper examines vertical dust fluxes from a portion of the Oceano Dunes in Coastal California under conditions of windblown transport. It reports on the results from a series of measurements conducted with optical particle counter-style instruments that were spaced vertically to allow for the calculation of dust flux. A second measurement component involved the mineralogical analysis of sand grains to infer the source material of emitted dust. These experiments were conducted simultaneously and in coordination with measurement of sand transport (reported separately). The authors

include a discussion of how their results may be applicable to global emissions from sand dunes.

Overall, this was a well written paper on an important topic. I recommend its publication following some revisions. There are some technical areas that ought to be addressed. Additionally, I found that the assertion that the results from this work are relevant for global emissions from sand dunes to be somewhat overstated. I believe this is a stylistics judgment, but have provided comments that the authors may wish to consider.

The authors rely on the concentration differences between Met One Particle Profiler instruments (model 212) to calculate the vertical flux of dust. If I read correctly, they basically use the two instruments that are mounted closest to the ground. As can be seen from the calibration coefficients in Table S2, these instruments are not exactly the gold standard when it comes to aerosol measurement. They are very useful for the type of field work described, but care should be taken in using the information quantitatively. The inter-instrument comparability changes over time as well as over the concentration range of the measurement. For example, two instruments might have one relationship when the (say) PM10 concentration is 50 micrograms per cubic meter and another relationship at 300 micrograms per meter cubed. In view of this, it would be important to ascertain that the concentration ranges that were experienced by the instruments during calibration and the ranges experienced during vertical flux measurement were similar. If not, this could be a significant source of bias in the results.

Another instrument-related observation is that the results and several of the key findings are related to the fact that the measured particle size distribution (PSD) during wind erosion events appears to become finer with increasing friction velocities. There is the potential that this is a consequence of measurement artifact associated with the changing aspiration efficiency of the inlet with wind speed. Typically, inlets to particulate matter instruments are designed to allow over a range of wind speeds for large particles (larger than the maximum size of interest) to make the treacherous aerody-
namic journey from being in the open air to passing by the actual sensing element, whatever that happens to be. The Met One instrument almost certainly loses particle collection efficiency at higher wind speeds. This loss of efficiency is most pronounced for larger (say > 5 microns) particles, giving the appearance that the size distribution is becoming finer at higher wind speeds. The authors should explicitly address/examine this issue and correct any of their results and subsequent conclusions as needed.

On a related point, I think the subtraction of the sea salt deposition profile is something of a distraction and perhaps also an artifact of the measurements. However, if the authors choose to retain this portion of the analysis, I would suggest first providing an estimate of the magnitude of sea salt deposition that is inferred and whether this number seems reasonable at all. Second, what mechanism of deposition could account for such deposition rates? Third, is that mechanism shear stress independent as is assumed in the final treatment of the numbers?

On a stylistic point, I can understand that there is value in expanding the results from the Oceano Dunes to try to understand dust emissions from sand dunes globally. While the authors are careful to caveat their comparisons between the dust emission schemes of (for example) the Sahara and Oceano Dunes, the language in which their work is described blurs out some of the caveats that are stated. For example, in the abstract, it states that "These measurements thus support the hypothesis that considerable emissions of fine dust can be generated by the reactivation of inactive dunes with accumulated clay minerals." Do they, actually? Elsewhere in the Abstract, there is the statement that "As such, dust emitted from sand sheet, and potentially from other active sands affected by similar dust emission processes, could have potent impacts on climate change, the hydrological cycle, and human health." Sure, this seems quite likely, but it follows a sentence that starts with "We further find...", giving the impression that the results of the paper pertain to climate, hydrological, and health impacts of dust from sand dunes. I have provided examples from the Abstract, but there are several locations in the manuscript where fuzzy sentence transitions amidst caveated

statements unintentionally inflate the achievements of the paper. Another example is the series of sentences on page 16 starting on line 4 with "we find that the PSD of dust..." through "...clay coating removal is likely a major emission process for active sands" on line 12. I would urge the authors to check the text for statements that may be misinterpreted as overreaching.

Minor comments: - The legends, axis titles, and figure titles of Fig S5 are too small to be legible. Suggest making more readable, especially if you elect to retain sea salt deposition corrections - Can you show the size distribution of Bullard et al Australian laboratory measurements alongside those presented in Figure 3? - Can you provide a brief explanation (few sentences describing main features of the technique) of how XRPD information is different from SEM-EDS and to what extent either provides bulk versus surface information? - Can you provide more detail on how many particles/samples (if applicable) were examined with XRPD and SEM/EDS? I couldn't find details of this work and it seems to be a key point of the paper that the feldspars and clay coatings are important. - There are likely gray literature reports where the dust emissions were measured at the Oceano Dunes. Naturally, methodology would not be exactly comparable, but it might be worthwhile to ask your ODSVRA contact to provide some of those reports for a gross comparison. Perhaps there is a journal policy on gray literature that precludes this.

---

## Referee Comment (RC2) · Anonymous Referee #2 · 10 Nov 2018

General comment: This study highlights the significance of sand dunes as an important source of fine dust to the global dust cycle. An in situ field measurement is undertaken at a costal location (Oceano Dunes) in California to understand the various processes involved in production of aeolian dust from active sands. A suite of instruments were installed in a vertical tower to monitor meteorological and physical parameter to get insight of the physics associated with dust emission from sands. In addition, chemical and mineralogical composition of sand from experimental site were also measured using XRD and SEM-EDS. Based on their result, authors have inferred two major mechanism for fine dust emission, (1) aeolian abrasion of feldspar and (2) removal of clay-mineral coating during saltation, from sands at Oceano site. These inferences are well supported by the previous experiments and literature. The manuscript is well

written, easy to read and of great significance for dust research. Thus, I recommend for publication in ACP, However, below I have made few comments which authors may think to consider while revising.

Specific comments:

Section 4.2: The discussion on representativeness of Oceano dust emissions can be improved further. As authors have tried to propose " . . . sand dunes might be globally relevant source of dust." (Page 1 line no. 14-15) and the feldspar abrasion is the major mechanism for such emission. They can include the distribution of feldspar content in active sand dunes and can make an approx. estimation of fine dust emitted by these source. It is important in order to link its contribution to global dust cycle.

Section 4.3: The implication of this study to human health and park manangement is OK, but stretching it to hydrological cycle and climate is overstatement. This study has not quantified feldspar content being emitted and subsequently carried to higher altitudes. So, their role in ice nuclei formation and consequently on the hydrological cycle is highly speculative. The Oceano dust emission might have a local to regional impact. However, the process of dust emission from sand dunes may have global relevance.

Section 4.4: I think authors should have attempted to collect aeolian dust samples on substrate using high volume samplers simultaneously with the other measurements. The enrichment of feldspar in the collected dust samples compared to parent sand had provided robust evidence for the proposed hypothesis of emission of fine dust from feldspar abrasion. This can be included in the limitation section.

Table 2: How many sand samples were analysed to get mineralogical composition? Mention in the table caption.

[Figure]

---

## Author Comment (AC1) · 17 Jan 2019

We thank both reviewers and the editor for their constructive comments, which has greatly helped us to improve the paper. Below we include a point-by point response to the referee comments, and describe the corresponding changes we have made to the manuscript.

V. Etyemezian (Referee #1) comments

This paper examines vertical dust fluxes from a portion of the Oceano Dunes in Coastal California under conditions of windblown transport. It reports on the results from a series of measurements conducted with optical particle counter-style instruments that were spaced vertically to allow for the calculation of dust flux. A second measurement component involved the mineralogical analysis of sand grains to infer the source material of emitted dust. These experiments were conducted simultaneously and in coordination with measurement of sand transport (reported separately). The authors include a discussion of how their results may be applicable to global emissions from sand dunes.

Overall, this was a well written paper on an important topic. I recommend its publication following some revisions. There are some technical areas that ought to be addressed. Additionally, I found that the assertion that the results from this work are relevant for global emissions from sand dunes to be somewhat overstated. I believe this is a stylistics judgement, but have provided comments that the authors may wish to consider.

We thank the reviewer for his constructive and helpful comments that helped us to further improve the paper. We address these comments below.

The authors rely on the concentration differences between Met One Particle Profiler instruments (model 212) to calculate the vertical flux of dust. If I read correctly, they basically use the two instruments that are mounted closest to the ground. As can be seen from the calibration coefficients in Table S2, these instruments are not exactly the gold standard when it comes to aerosol measurement. They are very useful for the type of field work described, but care should be taken in using the information quantitatively. The inner-instrument comparability changes over time as well as over the concentration range of the measurement. For example, two instruments might have one relationship when the (say) PM 10 concentration is 50 micrograms per cubic meter and another relationship at 300 micrograms per meter cubed. In view of this, it would be important to ascertain that the concentration ranges that were experienced by the instruments during calibration and the ranges experienced during vertical flux measurements were similar. If not, this could be a significant source of bias in the results.

Our figure Fig. S3A in the supplementary material shows that the calibration coefficients are nearly invariant with aerosol concentration; see also Table S2. Nonetheless, we accounted for the variation in the calibration coefficients with aerosol number concentration (see Table S2) and propagated this uncertainty throughout the analysis in the paper. To make this clearer, we added in line 24 of page 4 of the main text that (change underlined) "This procedure yielded a concentration-dependent correction factor with uncertainty for each size bin of each OPC (Table S2), which we propagated throughout our analysis (Fig. S3B)."

Another instrument-related observation is that the results and several of the key findings are related to the fact that the measured particle size distribution (PSD) during wind erosion events appears to become finer with increasing friction velocities. There is the potential that this is a consequence of measurement artifact associated with the changing aspiration efficiency of the inlet with wind speed. Typically, inlets to particulate matter instruments are designed to allow over a range of wind speeds for larger particles (larger than the maximum size of interest) to make the treacherous aerodynamic journey from being in the open air to passing by the actual sensing element, whatever that happens to be. The Met One instrument

almost certainly loses particle collection efficiency at higher wind spends. This loss of efficiency is most pronounced for larger (say >5 microns) particles, giving the appearance that the size distribution if becoming finer at higher wind speeds. The authors should explicitly address/examine this issue and correct any of their results and subsequent conclusions as needed.

We thank the reviewer for this valuable comment, which we now address in the revised version. Unfortunately, the manufacturer of Met One instruments could not share a curve of the dependence of sampling efficiency (i.e., the ratio of particle concentration measured by the sensor to the particle concentration in the ambient air; Von der Weiden et al., 2009) on wind speed and particle size. On the other hand, the study of Von der Weiden et al. (2009) obtained general results on the sampling efficiency as a function of particle size, wind speed, and sensor inlet characteristics. This work supports the reviewer's argument that the efficiency of collection decreases strongly with wind speed for coarse particles, for instance the loss rate can approach 100% for particles larger than 10 μm in diameter under strong wind events. Therefore, we have removed the results of our largest bin (Bin 7 with the nominal diameter of 10 μm) (see line 31 in page 3, lines 1-5 in page 4, lines 16-23 in page 8, Figs. 3, 4, and 6) and have added a discussion in the limitation section of how the possible decrease of the sampling efficiency for the other bins (notably Bin 6) affects our result that the PSD of emitted dust remains variant with shear velocity in Fig. 4 (see lines 15-19 in page 14).

On a related point, I think the subtraction of the sea salt deposition profile is something of a distraction and perhaps also an artifact of the measurements. However, if the authors choose to retain this portion of the analysis, I would suggest first providing an estimate of the magnitude of sea salt deposition that is inferred and whether this number seems reasonable at all. Second, what mechanism of deposition could account for such deposition rates? Third, is that mechanism shear stress independent as is assumed in the final treatment of the numbers?

This is a good comment. We have added the magnitude of sea salt deposition flux (see line 4 in page 7) and validated against past published study on coastal sea salt deposition (see lines 5-6 in page 7). However, it remains difficult to address the second and third points, because of the complexity of coastal sea salt aerosols. Sea salt emission from coastal ocean increases with shear velocity (O'Dowd and de Leeuw, 2007), and the deposition changes downwind with shear velocity and distance to the shoreline. The sum of those effects depends on a variety of parameters which were not measured in our dataset. On the other hand, based on our available data, we could clearly see the signal of sea salt deposition when saltation is inactive (see lines 24-29 in page 6). Because the assumption of sea salt deposition being invariant with shear velocity leads to the same PSD as the assumption of sea salt deposition flux increasing with shear velocity (see Fig. S9), we consider the effect of sea salt deposition to be of second order in our measurements.

On a stylistic point, I can understand that there is a value in expanding the results from the Oceano Dunes to try to understand dust emissions from sand dunes globally. While the authors are careful to caveat their comparisons between the dust emission schemes of (for example) the Sahara and Oceano Dunes, the language in which their work is described blurs out some of the caveats that are stated. For example, in the abstract, it states that "These measurements thus support the hypothesis that considerable emissions of fine dust can be generated by the reactivation of inactive dunes with accumulated clay minerals." Do they, actually?

Elsewhere in the Abstract, these is the statement that "As such, dust emitted from sand sheet, and potentially from other active sands affected by similar dust emission processes, could have potent impacts on climate change, the hydrological cycle, and human health," Sure, this seems quite likely, but it follows

a sentence that starts with "We further fund…", giving the impression that the results of the paper pertain to climate, hydrological, and health impacts of dust from sand dunes.

I have provided examples from the Abstract, but there are several locations in the manuscript where fuzzy sentence transitions amidst caveated statements unintentionally inflate the achievement of the paper. Another example is the series of sentence of page 16 starting on line 4 with "we find that the PSD of dust…" through "… clay coating removal is likely a major emission process for active sands" on line 12. I would urge the authors to check the text for statements that may be misinterpreted as overreaching.

We thank the reviewer for his helpful suggestion. We have revised the abstract and removed "These measurements thus support the hypothesis that considerable emissions of fine dust can be generated by the reactivation of inactive dunes with accumulated clay minerals." from the abstract (see lines 11-20 in page 1). Furthermore, we have systematically rewritten the discussion section, especially section 4.2 about the representativeness of dust emission from Oceano (see lines 12-31 in page 11 and lines 1-14 in page 12) and section 4.3 about the implications for downwind human health, park management, hydrological cycle, and climate (see lines 15-31 in page 12, lines 1-32 in page 13, and lines 1-3 in page 14).

Minor comments: - The legends, axis titles, and figure titles of Fig. S5 are too small to be legible. Suggest making more readable, especially if you elect to retain sea salt deposition corrections.

We have enlarged Fig. S5 in the revised manuscript, detailed in pages 17-18 of the Supplement.

Can you show the size distribution of Bullard et al. Australian laboratory measurements alongside those presented in Figure 3?

We have included Fig. S10 in the revised Supplement that shows the PSD of produced dust by clay-coating removal (Bullard et al., 2004). We chose not to include Bullard et al. (2004)'s results in the main text, because their measured size distributions differ greatly between different times of measurement in their abrasion apparatus (see their Fig. 17) such that no clear conclusions can be drawn from the comparison.

Can you provide a brief explanation (few sentences describing main features of the technique) of how XRPD information is different from SEM-EDS and to what extent either provides bulk versus surface information?

The XRPD method uses data on the crystal structure to retrieve the minerals that make up the bulk sample. On the other hand, SEM-EDS provides information on the coating minerals. We have re-written the final paragraph of section 2 (lines 9-29, page 7) to make this clearer.

Can you provide more detail on how many particles/samples (if applicable) were examined with XRPD and SEM/EDS? I couldn't find details of this work and it seems to be a key point of the paper that the feldspars and clay coatings are important.

We collected 2 soil samples (each ~220 grams) from the tower location and 100 meters upwind, respectively. Four 1-gram samples were examined with XRPD (add on Table 2's caption). Six 1-gram samples were examined with SEM/EDS. We have re-written the final paragraph of section 2 (lines 9-29, page 7) to include these details.

There are likely gray literature reports where the dust emissions were measured at the Oceano Dunes. Naturally, methodology would not be exactly comparable, but it might be worthwhile to ask your ODSVRA contact to provide some of those reports for a gross comparison. Perhaps there is a journal policy on gray literature that precludes this.

Thank you for pointing this out. We checked with our Oceano contact (Jack Gillies), who could not point out any relevant gray literature but instead suggested we add the following reference: Gillies, J.A., Etyemezian, V., Nikolich, G., Glick, R., Rowland, P., Pesce, T., Skinner, M., 2017. Effectiveness of an array of porous fences to reduce sand flux: Oceano Dunes, Oceano CA. Journal of Wind Engineering and Industrial Aerodynamics 168, 247–259. https://doi.org/10.1016/j.jweia.2017.06.015. We now refer to this paper as an example of dust emission concentrations measured in the riding area of Oceano Dunes SVRA (lines 28-31 of page 12). We do not know of any published direct comparisons between dust emissions from riding versus non-riding areas at Oceano SVRA.

Anonymous Referee #2 comments

General comments: This study highlights the significance of sand dunes as an important source of fine dust to the global dust cycle. An in situ field measurement is undertaken at a costal location (Oceanic Dunes) in California to understand the various processes involved in production of aeolian dust from active sands. A suite of instruments were installed in a vertical tower to monitor meteorological and physical parameter to get insight of the physics associated with dust emission from sands. In addition, chemical and mineralogical composition of sand from experimental site were also measured using XRD and SEM-EDS. Based on their result, authors have inferred two major mechanism for fine dust emission, (1) aeolian abrasion of feldspar and (2) removal of clay-mineral coating during saltation, from sands at Oceano site. These inferences are well supported by the previous experiments and literature. The manuscript is well written, easy to read and of great significance for dust research. Thus, I recommend for publication in ACP, however, below I have made few comments which authors may think to consider while revising.

Thank you for the positive comments. We address your comments carefully as follows.

Specific comments:

Section 4.2: The discussion on representativeness of Oceano dust emissions can be improved further. As authors have tried to propose "… sand dunes might be globally relevant source of dust." (Page 1 line no. 14-15) and the feldspar abrasion is the major mechanism for such emission. They can include the distribution of feldspar content in active sand dunes and can make an approx. estimation of fine dust emitted by these sources. It is important in order to link its contribution to global dust cycle.

This is a very good suggestion. However, our results are based on only one field site and we cannot separate the effects of emission through clay-coating removal and feldspar abrasion. We thus think that extrapolating our results to estimate dust emissions through feldspar abrasion on a global scale, though clearly desirable, would require substantial future measurements at a range of sand dunes with different properties (including sand size distribution, mineralogy (such as extents of clay-mineral coatings and feldspars), chemical and physical weathering rates, dune type, and palaeoenvironmental history).

Section 4.3: The implication of this study to human health and park management is OK, but stretching it to hydrological cycle and climate is overstatement. This study has not quantified feldspar content being emitted and subsequently carried to higher altitude. So, their role in ice nuclei formation and consequently on the hydrological cycle is highly speculative. The Oceano dust emission might have a local to regional impact. However, the process of dust emission from sand dunes may have global relevance.

We thank the reviewer for these helpful comments. We have re-written section 4.3 about the implications for downwind human health, park management, hydrological cycle, and climate (see lines 15-31 in page 12, lines 1-32 in page 13, and lines 1-3 in page 14) accordingly, and now point out that the impact of Oceano

dust on downwind cloud microphysics remains speculative. We now focused this paragraph more on stressing the need for further research to investigate links of feldspar emission to the hydrological cycle.

Section 4.4: I think authors should have attempted to collect aeolian dust samples on substrate using high volume samplers simultaneously with the other measurements. The enrichment of feldspar in the collected dust samples compared to parent sand had provided robust evidence for the proposed hypothesis of emission of fine dust from feldspar abrasion. This can be included in the limitation section.

Thank you for this suggestion. We now include this as limitation #2 in section 4.4 (see line 8 in page 14).

Table 2: How many sand samples were analyzed to get mineralogical composition? Mention in the table caption.

We collected 4 sand samples (each ~220 grams) in total with 2 samples from the tower location and 2 from 100 meters upwind. For each of the 4 samples, we used 1 gram for XRPD analysis to obtain mineralogical composition of bulk sand grains. We have detailed in Table 2's caption and re-written the final paragraph of section 2 (lines 9-29, page 7) to make this clearer.